# Risk of Reflux-Related Symptoms and Reflux Esophagitis after *Helicobacter pylori* Eradication Treatment in the Japanese Population

**DOI:** 10.3390/jcm10071434

**Published:** 2021-04-01

**Authors:** Mitsushige Sugimoto, Masaki Murata, Eri Iwata, Naoyoshi Nagata, Takao Itoi, Takashi Kawai

**Affiliations:** 1Department of Gastroenterological Endoscopy, Tokyo Medical University Hospital, Shinjuku, Tokyo 160-0023, Japan; eiwata@tokyo-med.ac.jp (E.I.); nnagata_ncgm@yahoo.co.jp (N.N.); t-kawai@tokyo-med.ac.jp (T.K.); 2Division of Digestive Endoscopy, Shiga University of Medical Science Hospital, Otsu 520-2192, Japan; 3Department of Gastroenterology, National Hospital Organization Kyoto Medical Center, Kyoto 612-8555, Japan; mura05310531@gmail.com; 4Department of Gastroenterology and Hepatology, Tokyo Medical University Hospital, Shinjuku, Tokyo 160-0023, Japan; itoitakao@gmail.com

**Keywords:** GERD, *Helicobacter pylori*, eradication therapy, gastric acid, questionnaires

## Abstract

Backgrounds: A meta-analysis of reports primarily from Western countries showed no association between *Helicobacter pylori* eradication and reflux esophagitis development. The risk of reflux esophagitis may differ among different populations based on *H. pylori* virulence factors and acid secretion ability. We evaluated the prevalence rates of reflux esophagitis in *H.-pylori*-positive Japanese subjects and assessed risk factors for reflux esophagitis after eradication. Methods: Among 148 *H.-pylori*-positive subjects who underwent *H. pylori* eradication from August 2015 to December 2019, we evaluated the prevalence of reflux esophagitis on endoscopy at 12 months after eradication success and the severity of reflux-related symptoms by the F-scale questionnaire at 2 months after treatment and 12 months after eradication success. Results: The prevalence of reflux esophagitis in *H.-pylori*-positive patients at entry was 2.0% (3/148). At 12 months after eradication success, the prevalence was 10.8% (16/148) (*p* < 0.01). In the F scale, the median total score before treatment was 4 (range: 0–49), which significantly decreased to 2 (range: 0–22) (*p* < 0.01) at 2 months after treatment and 3 (range: 0–23) (*p* < 0.01) at 12 months after eradication success. Following multivariate analysis, the pretreatment total F-scale score was a risk factor for the development of reflux esophagitis (odds ratio: 1.069, 95% confidence interval: 1.003–1.139, *p* < 0.01). Conclusions: In this *H.-pylori*-positive Japanese population, eradication therapy was associated with reflux esophagitis in around 10% of patients, particularly in those with severe reflux-related symptoms at baseline. Reflux-related symptoms may improve throughout the 12 months after successful eradication therapy, irrespective of the development of reflux esophagitis.

## 1. Introduction

Gastroesophageal reflux disease (GERD) is one of the most common upper gastrointestinal diseases worldwide [1]. In Japan, acid-reflux-related symptoms are mild and less frequent than in Western populations [2]. In general, although the pathogenesis of GERD is multifactorial (i.e., frequent and prolonged reflux of gastric acid, status and size of hiatal hernia, severity of esophageal sphincter dysfunction, decrease in esophageal motility dysfunction, increase in hypersensitivity, and status of *Helicobacter pylori* infection) [3], recent attention has focused on the association of *H. pylori* with the development of reflux esophagitis [4,5]. In particular, long-term *H. pylori* infection decreases the stomach’s ability to secrete gastric acid via the progression of gastric mucosal atrophy and exacerbation of gastric mucosal inflammation [6], and *H. pylori* infection is therefore inversely associated with the development of reflux esophagitis [7,8,9]. Because the incidence of reflux esophagitis differs among different populations, however, it is unknown whether the association of reflux esophagitis with *H. pylori* is similar among different populations.

Although eradication therapy for *H. pylori* infection reduces the risk of gastric cancer development and esophageal cancer, consistent with the *H. pylori* protective theory [10,11], it is unclear whether *H. pylori* eradication increases the risk of reflux esophagitis. Eradication therapy for *H. pylori* infection is considered to increase the severity of esophageal damage and reflux-related symptoms in patients with reflux esophagitis, and after eradication, around 10% of patients experience reflux-related symptoms, irrespective of their experience with reflux-related symptoms before eradication therapy [12]. However, a recent meta-analysis using reports primarily from Western countries showed no association between *H pylori* eradication and reflux esophagitis [4,5,13,14,15]. Because only a few reports from Japan have investigated reflux esophagitis after eradication [16], and given that the genetic, social, and bacterial background related to acid secretion differs between Western and East Asian populations [17,18], a study focused on a Japanese population may be valuable in clarifying associations with *H. pylori* infection, acid secretion, reflux esophagitis, and reflux-related symptoms.

Accordingly, the aims of this study were: (1) to determine the prevalence of reflux esophagitis and reflux-related symptoms in *H.-pylori*-positive Japanese patients; (2) to evaluate the association between *H. pylori* eradication and reflux esophagitis in the Japanese population; and (3) to investigate the risk of reflux esophagitis after eradication in the Japanese population.

## 2. Materials and Methods

### 2.1. Patients and Study Protocol

The protocol of this study was reviewed and approved by the Institutional Review Board of Shiga University of Medical Science. This study enrolled 148 *H.-pylori*-positive patients from August 2015 to December 2019 (Table 1). Inclusion criteria were age ≥20 years with *H. pylori* infection, no medication by vonoprazan, proton pump inhibitor (PPI) and histamine 2 receptor antagonist (H2RA), performance of endoscopy to evaluate reflux esophagitis before eradication therapy and at 12 months after eradication success, use of a questionnaire to evaluate reflux-related symptoms before eradication therapy and at 2 months after treatment and 12 months after eradication success in our University Hospital, and conclusive evaluation of eradication outcome by the ^13^C-urea breath test (UBIT 100 mg tablets, Otsuka Pharmaceutical Co., Ltd., Tokyo, Japan, using a cut-off of 2.5‰). Because this study was conducted under a retrospective observational design and written informed consent was not obtained from each enrolled patient, a document that reported an opt-out policy by which potential patients and/or relatives could refuse inclusion was uploaded on the web page of Shiga University of Medical Science Hospital. The study protocol conformed to the ethical guidelines of the Declaration of Helsinki [19]. 

*H. pylori* infection was diagnosed in all patients using the rapid urease test (Helicocheck^®^; Institute of Immunology, Co., Ltd., Tochigi, Japan) and a culture test (BML, Inc., Tokyo, Japan). Patients were diagnosed *H. pylori* infection positive if at least one of the two tests was positive. 

For bacterial culture and antimicrobial sensitivity testing, agar plates were inoculated with biopsy specimens and incubated at 37 °C under microaerophilic conditions (5% O_2_, 10% CO_2_, and 85% N_2_) for approximately 7 days at 37 °C. *H. pylori* was identified using oxidase production. *H. pylori* colonies were subcultured using the agar dilution method to determine the minimum inhibitory concentration (MIC) for amoxicillin, metronidazole, clarithromycin, and sitafloxacin, according to the recommendations of the Clinical and Laboratory Standards Institute (CLSI) [20] and the manufacturer’s instructions. Cut-off MICs used to define resistance were >1.0 µg/mL for clarithromycin and sitafloxacin and >8 µg/mL for metronidazole [21,22,23]. For amoxicillin, the cut-off MICs used to define resistance and the absence of sensitivity were >0.5 and >0.06 µg/mL, respectively.

All *H.-pylori*-positive patients underwent gastroduodenal endoscopy for evaluation of the presence of reflux esophagitis, Barrett’s esophagus, and hiatal hernia, as well as the endoscopic severity of gastritis. Patients infected with *H. pylori* received eradication treatment, as below. At 6 to 8 weeks after eradication treatment, success was evaluated by the ^13^C-urea breath test with a cut-off value of 2.5 ‰. At 12 months after eradication success, the development of reflux esophagitis was evaluated by endoscopy. It is usually recommended to undergo a second endoscopy after 12 months for patients by the guidelines for the management of *Helicobacter pylori* infection in Japan [24]. Patients were also evaluated at 2 months after treatment and 12 months after eradication success for the severity of abdominal symptoms using the F scale [25,26]. Significant differences in reflux-related symptoms before eradication therapy and at 2 months after treatment and 12 months after eradication success were evaluated. Patients were not evaluated at 12 months after eradication success for the presence of *H. pylori* by the urea breath test or other tests.

### 2.2. H. pylori Eradication Therapy 

In Japan, as the standard of care, *H. pylori* eradication therapies are currently limited to regimens comprising acid-inhibitory drugs such as a PPI or vonoprazan, amoxicillin, and clarithromycin for 7 days as a first-line eradication regimen, as well as PPI or vonoprazan, amoxicillin, and metronidazole for 7 days as a second-line eradication regimen [24]. Given that a recent meta-analysis showed greater efficacy for vonoprazan-containing regimens compared with a PPI-containing regimen [27], all patients in the present study underwent eradication with vonoprazan 20 mg twice-daily dosing (bid) and a combination of two antibiotics, namely clarithromycin (200 mg bid) and amoxicillin (750 mg bid) as first-line eradication treatment (*n* = 107), metronidazole (250 mg bid) and amoxicillin (750 mg bid) as second-line treatment (*n* = 26), and sitafloxacin (100 mg bid) and amoxicillin (500 mg bid) as third-line treatment (*n* = 15), all for 7 days. When initial eradication therapy failed, patients received advanced eradication therapy, namely a second-line eradication regimen for patients who failed a first-line regimen, a third-line regimen for patients who failed a second-line regimen, and a fourth-line regimen (sitafloxacin (100 mg bid) and metronidazole (250 mg bid)) for those who failed a third-line regimen.

### 2.3. Endoscopy and Severity of Gastritis

Reflux esophagitis was assessed according to the Los Angeles classification (grades A to D) [28]. In addition, redness was endoscopically defined as mucosal findings of redness, edema, or white granules in the EG junction, irrespective of the presence of reflux-related symptoms. GERD was defined as reflux esophagitis or redness. 

Severity of gastritis was evaluated using the Kyoto classification [29,30]. Barrett’s esophagus was diagnosed endoscopically if columnar-appearing mucosa was observed between the squamocolumnar and EG junction. Hiatal hernia was diagnosed when greater than 2 cm dislocation of EG junction toward the esophageal site was found endoscopically [31].

### 2.4. Data Analysis 

Values for age, height, and body weight are given as the mean ± standard deviation (S.D.). Scores for the Endoscopic Kyoto classification and the questionnaire are given as the median and range. The eradication rate of *H. pylori* was evaluated by intention-to-treat (ITT) analysis and calculated with 95% confidence intervals (CIs). Statistically significant differences in endoscopic scores and symptom scores among the three groups (non-GERD, redness, and reflux esophagitis) were determined by the Mann–Whitney U test when significant differences were observed by the Kruskal–Wallis test. To determine whether endoscopic and symptom scores differed among the observational time points (0, 2, and 12 months), the Wilcoxon signed-rank test was used. Statistically significant differences in mean values of age, height, and body weight among the three groups (non-GERD, redness, and reflux esophagitis) were determined by one-way ANOVA followed by the Scheffé multiple comparisons test. Statistically significant differences in category data among the three groups were determined by the χ^2^ test. Univariate and multivariate logistic regression analyses were used to test the associations of 16 candidate variables with the development of reflux esophagitis. Multicollinearity among the variables was tested using the variance inflation factor (VIF). The multivariate analysis examined the risk of reflux esophagitis using factors that showed *p* < 0.2 in the univariate analysis with adjustment for age and sex. A value of *p* < 0.05 was considered statistically significant and all *p*-values were two-sided. Calculations were conducted using SPSS version 20 (IBM Inc.; Armonk, NY, USA).

The sample size and power were calculated by the *t*-test and set such that the effects size was 0.3, the correlation coefficient was 0.6, the desired power was 80%, with a significance level of 0.05 in a two-sided test, and the required sample number was 134. In addition, we expected that 10% of patients enrolled in the study would delete by loss of data; we therefore aimed to enter 148 patients for a valid analysis.

## 3. Results

### 3.1. Patient Characteristics

Of the 148 *H.-pylori*-positive Japanese patients, the prevalence of reflux esophagitis findings on endoscopy at baseline was 2.0% (3/148), and a total of 21.6% (32/148) had redness (Table 1). Most characteristics and endoscopic findings were similar among the redness, reflux esophagitis, and non-GERD groups (Table 1). In addition, in the F-scale questionnaire, all scores in *H.-pylori*-positive patients for redness, reflux esophagitis, and the non-GERD groups at baseline were similar (Table 1). 

Eradication rates in the ITT analysis were 85.0% (95% CI: 76.9–91.2%, 91/107) for first-line therapy, 88.5% (95% CI: 69.8–97.6%, 23/26) for second-line therapy, and 93.3% (95% CI: 68.1–99.8%, 14/15) for third-line therapy. Twenty patients who failed initial eradication therapy received advanced eradication therapy, which was successful in all patients.

### 3.2. Endoscopic Reflux Esophagitis after H. pylori Eradication Therapy

At 12 months after eradication success, esophagitis was evaluated using endoscopy. The prevalence of GERD was 10.8% for reflux esophagitis (16/148), including 6.8% GERD grade A (10/148), 4.1% grade B (6/148), and 17.6% redness (26/148) (Table 2). The prevalence of reflux esophagitis differed between baseline and after treatment (2.0% (3/148) vs. 10.8% (16/148), *p* < 0.01). Reflux esophagitis developed *de novo* in 7.1% (8/113) in the non-GERD group at pretreatment and in 18.8% (6/32) in the redness group (Table 3).

There was no significant difference in rates of hiatus hernia and SSBE, or in the severity of endoscopic atrophy and intestinal metaplasia, among the redness, reflux esophagitis, and non-GERD groups at 12 months after eradication success (Table 2). For the F scale, the acid reflux, dysmotility-related, and total scores in the reflux esophagitis group were higher than those in the non-GERD group (Table 2). 

### 3.3. Symptomatic Reflux Esophagitis after H. pylori Eradication Therapy

At 2 months after treatment and 12 months after eradication success, acid-reflux-related symptoms were evaluated. The acid reflux, dysmotility-related, and total scores in the F scale were significantly decreased from scores at baseline (median total score of the F-scale questionnaire: 4 (range: 0–49) at baseline, 2 (range: 0–22) at 2 months after treatment (*p* < 0.01), and 3 (range: 0–23) at 12 months after eradication success (*p* < 0.01)) (Figure 1). 

### 3.4. Reflux Esophagitis after H. pylori Eradication and Outcome of Eradication Therapy

The total score for the F-scale questionnaire at 2 months after treatment showed significant differences between patients with successful (*n* = 128) and failed treatment (*n* = 20) (*p* = 0.04) (Figure 2). Acid reflux scores, dysmotility-related scores, and total scores in patients with failed treatment did not differ between pretreatment and after treatment, while scores in patients with successful treatment decreased (Figure 2).

### 3.5. Time Course of F-Scale Questionnaire Scores between Patients with Non-Erosive and Reflux Esophagitis

When we compared the scores for the F-scale questionnaire in patients with reflux esophagitis and patients with redness, baseline scores were similar between the two groups. The acid reflux, dysmotility-related, and total scores significantly differed between the two groups at 2 months after treatment (*p* < 0.01, 0.02 and <0.01, respectively) and 12 months after eradication success (*p* < 0.01, <0.01, and <0.01) (Figure 3). The acid reflux, dysmotility-related, and total scores in patients with no erosive esophagitis significantly decreased from pretreatment to 2 months after treatment and 12 months after eradication success. In contrast, scores in patients with reflux esophagitis showed no remarkable changes between pretreatment and 2 months after treatment and 12 months after eradication success (Figure 3).

We evaluated associations with scores of the F-scale questionnaire and different categories using sex, SSBE, and hiatal hernia (Table 4). The acid-related and total F-scale scores at entry (before eradication therapy) significantly differed among groups based on sex, SSBE, and hiatus hernia (*p* < 0.01 and *p* = 0.03, respectively). The F-scale scores in patients with both SSBE and hiatus hernia were higher than SSBE-negative, hiatal-hernia-negative, or both-negative patients, irrespective of sex, both before eradication therapy and at 12 months after eradication success.

### 3.6. Risk Factors for Reflux Esophagitis after H. pylori Eradication Therapy 

On univariate analysis, risk factors for reflux esophagitis development were the total baseline F-scale score (OR: 1.069, 95% CI: 1.009–1.132, *p* = 0.02) and the dysmotility-related score of the F scale at pretreatment (1.200, 1.055–1.364, *p* < 0.01) (Table 5). Following multivariate analysis using factors showing *p* < 0.2 in the univariate analysis (hiatal hernia and total score of F scale) and age and sex as adjustment factors, the total baseline F-scale score was again shown to be a significant risk factor (OR: 1.069, 95% CI: 1.003–1.139, *p* = 0.04) (Table 5).

## 4. Discussion

We investigated the association between *H. pylori* eradication therapy and reflux esophagitis development in Japanese subjects, who are generally at lower risk of reflux esophagitis than Western populations. This *H.-pylori*-positive cohort had a GERD rate of 23.6%, including 2.0% with reflux esophagitis and 21.6% with redness. In addition, although *H. pylori* eradication may affect the severity of reflux esophagitis and reflux-related symptoms, and *de novo* reflux esophagitis was revealed in 4.8–20.5% of Japanese *H.-pylori*-positive patients after eradication [12], the prevalence of reflux esophagitis at 12 months after eradication success was 10.8%. On the other hand, questionnaires revealed that reflux-related symptoms were improved at 2 months after treatment and 12 months after eradication success, especially in patients with successful treatment and patients without GERD or reflux esophagitis. 

In a meta-analysis of cohort studies, the prevalence of *H. pylori* in GERD patients was 38.2%, which was lower than that in patients without GERD (49.5%, OR: 0.58) [32]. In general, while GERD results from abnormal transient lower esophageal sphincter relaxation (TLESR) and an imbalance between esophageal mucosa exposure to acid and clearance mechanisms [33,34], the acidity of gastric juice, volume of gastric juice, and frequency of acid reflux into the esophagus also plays a role in the development of reflux esophagitis and symptoms. *H. pylori* is similarly known to potently inhibit gastric acid secretion through progressive atrophic changes in acid-producing gastric mucosal cells and the infiltration of activated inflammatory cells that secrete proinflammatory cytokines (e.g., IL-1β and TNF-α) [6]. In addition, as host genetic factors, the *TNF-A*, *IL-1B,* and *IL-1RN* genetic polymorphisms, which influence serum and gastric mucosal TNF-α and IL-1β levels, are inversely associated with the risk of reflux esophagitis development in *H.-pylori*-positive patients because their specific genotypes (e.g., *IL-1B*-511 T/T, *IL-1RN* *2/*2, *TNF-A*-857 T/T, -863 A/A, and -1031 C/C types) are linked to severe gastric mucosal atrophy, development of peptic ulcers and gastric cancer, and hypochlorhydria [6]. Infection with *H. pylori* strains with high virulence factors induces a more severe degree of gastric mucosal inflammation with hypochlorhydria. Epidemiological studies show that infection with the *H. pylori cagA*-positive strain is strongly negatively related to the development of reflux esophagitis [35]. However, because the infection rate of *H. pylori* is decreasing year by year, the prevalence of reflux esophagitis is expected to increase. 

Many studies have evaluated the effects of *H. pylori* eradication on the development of reflux esophagitis and/or GERD, but results have been inconsistent and inconclusive [5]. The Maastricht V Consensus Report recommended that *H. pylori* eradication therapy should not exacerbate pre-existing reflux esophagitis or affect treatment efficacy [36]. A recent meta-analysis evaluated whether eradication therapy affects the prevalence of GERD or reflux esophagitis and the severity of GERD or reflux esophagitis and found no significant difference in the prevalence of reflux esophagitis and/or GERD after eradication between patients with successful and unsuccessful eradication, irrespective of follow-up period after treatment or the presence or absence of baseline disease [5]. The frequency of GERD or reflux esophagitis was similar between patients with successful and unsuccessful eradication at 6 months (OR: 1.85, 95% CI: 0.68–5.04; *p* = 0.23) and 12 months (OR: 0.99, 95% CI: 0.64–1.52; *p* = 0.97) [5]. However, most of the studies included in this meta-analysis were from Europe and North America. Schwizer et al. [37] reported that eradication therapy did not result in the recovery of gastric acid secretion after eradication therapy in Western populations. In our meta-analysis, which focused on reflux esophagitis with endoscopic mucosal injury, we showed that incidences of *de novo* reflux esophagitis in Western and East Asian populations in the eradication group were 9.1% (132/1444; control noneradication group, 4.5%, 53/1176) and 21.2% (324/1530; control noneradication group, 10.7%, 48/447), respectively [38]. An East Asian report [39] showed that the total percentage of time at pH < 2 (2.1 ± 0.5 vs. 0.8 ± 0.2) increased in the eradication group compared with the noneradication group. Recently, a large study of 10,102 *H.-pylori*-positive Korean patients revealed that eradication therapy increased the prevalence of reflux esophagitis to 4.9% (490/10,102) [40]. In the present study, endoscopy showed a 10.8% rate of reflux esophagitis (16/148) at 12 months after eradication success. These differing results for the development of GERD or reflux esophagitis after eradication between Western and East Asian populations may be ascribable to differences in lifestyle, genetic factors, and/or virulence factors of *H. pylori* strains. 

In a previous meta-analysis, no significant differences were observed in the incidence of “heartburn” between *H.-pylori* eradicated patients and infected patients [38]. In this study, however, reflux-related symptoms evaluated by the F-scale questionnaire (combined scores of seven questions) significantly improved from baseline in all patients (Figure 1). In double-blind RCTs, *H. pylori* eradication decreased the severity of reflux-related symptoms in patients with duodenal ulcers [41]. Saad et al. [13] reported a significantly lower prevalence of reflux-related symptoms in the eradication group (13.8%) than the noneradication group (24.9%) (OR: 0.55, 95% CI: 0.35–0.87). However, it is unknown why reflux-related symptoms improve after eradication in East Asian populations with risk of reflux esophagitis after eradication. In this study, reflux-related symptoms in patients with GERD showed no remarkable changes throughout the observation period, as shown in Figure 3. This observation suggests that reflux-related symptoms in *H.-pylori*-positive symptomatic patients without GERD improved after eradication therapy and that most reflux-related symptoms on the F scale may be considered to be *H. pylori* infection-related abdominal symptoms, namely *H.-pylori*-associated dyspepsia. A meta-analysis reported that eradication therapy was effective in approximately 10% of functional dyspepsia patients, especially in Asian populations [42]. In addition, the Kyoto Global Consensus Meeting recently defined *H.-pylori*-associated dyspepsia as a condition in which abdominal symptoms disappeared or improved 6–12 months after eradication treatment [43]. On this basis, the improvement in abdominal symptoms after eradication in *H.-pylori*-positive patients might be attributable to improved *H.-pylori*-related and functional dyspepsia-related symptoms rather than to GERD-related symptoms, irrespective of the recovery of acid secretory ability after eradication therapy. We consider that it would be better to investigate the association of outcome of eradication therapy and improvement in reflux-related symptoms using 24 h intragastric and intraesophageal pH monitoring, as a further study.

Antimicrobial Stewardship Team (AST)-orientated treatment in Japan is major for most infectious diseases. In Japan, however, *H. pylori* eradication therapies are currently limited by the Japanese insurance system to regimens comprising an acid-inhibitory drug bid, amoxicillin 750 mg bid, and clarithromycin 200 mg or 400 mg bid for 7 days as a first-line eradication regimen, as well as PPI or VPZ bid, amoxicillin 750 mg bid, and metronidazole 250 mg bid for 7 days as a second-line eradication regimen, irrespective of the infection of resistant strains to antimicrobial agents [24]. Therefore, AST-orientated treatment is not given for *H. pylori* infection in Japan. I think that susceptibility-based tailored treatment should be selected for *H. pylori* eradication therapies in Japan.

This study has a few limitations. First, this is a single-center retrospective study with a small sample number. Second, although reflux esophagitis has multifactorial pathogenesis related to gastric acid and esophageal dysfunction and is influenced by the intake of PPI/vonoprazan and other medications (nonsteroidal anti-inflammatory drugs, aspirin, and calcium channel blockers), we had no data on intake of medications. Third, because this study did not enroll *H.-pylori*-positive patients who did not receive *H. pylori* eradication therapy, it is unclear whether the rate of de novo reflux esophagitis after eradication therapy is high or not. Fourth, because we did not limit the medication of acid secretion inhibitors after *H. pylori* eradication therapy, two patients received additional PPI therapy for reflux-related abdominal symptoms after successful *H. pylori* eradication therapy. This is considered to be a major methodological problem. Fifth, the sample size of this study is not great, and because this study was performed in Japan, as a single-center retrospective study, this study might have a bias of diagnosis/association for esophagitis

## 5. Conclusions

We found that the prevalence of erosive gastritis in Japanese *H.-pylori*-positive patients is <3% and that eradication for *H. pylori* infection is a risk factor for the *de novo* development of endoscopic reflux esophagitis, especially in patients with severe abdominal symptoms at baseline. However, eradication improved symptoms over a long period. Because *H. pylori* eradication effectively reduces the risk of gastric cancer development, irrespective of a past history of previous cancer, we recommend eradication therapy for *H. pylori* infection, as currently suggested by the treatment guidelines [36]. A comprehensive investigation of the development of reflux esophagitis and reflux-related symptoms after eradication therapy will be aided by prospective randomized trials enrolling both Western and East Asian populations (eradication group vs. placebo group) and considered confounding factors.

## Figures and Tables

**Figure 1 jcm-10-01434-f001:**
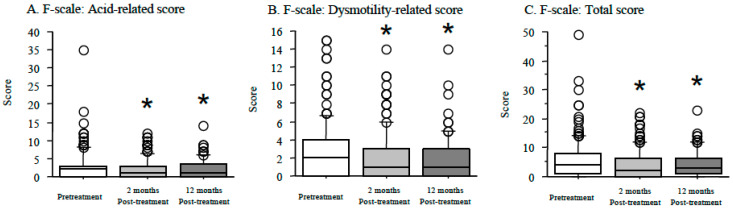
Association with the F scale (**A**–**C**) and time course at baseline, 2 months after treatment, and 12 months eradication success in 148 patients with successful *H. pylori* eradication evaluated in our hospital at 12 months after eradication success. Acid reflux, dysmotility-related, and total scores in the F-scale questionnaire were significantly decreased from scores at baseline. *: *p* < 0.05 vs. score at pretreatment.

**Figure 2 jcm-10-01434-f002:**
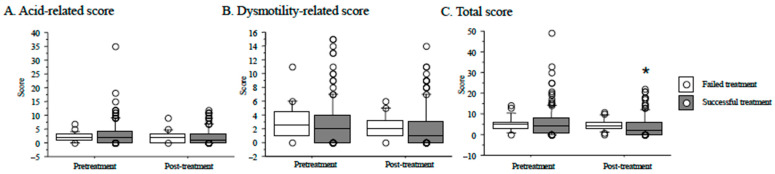
Correlation of the F-scale questionnaire score and time course from baseline to 2 months after treatment between patients with successful treatment and failed treatment in all 148 patients with successful *H. pylori* eradication. Scores in the F-scale questionnaire are acid-related (**A**), dysmotility-related (**B**), and total scores (**C**). *: *p* < 0.05 vs. score of patients with successful treatment.

**Figure 3 jcm-10-01434-f003:**
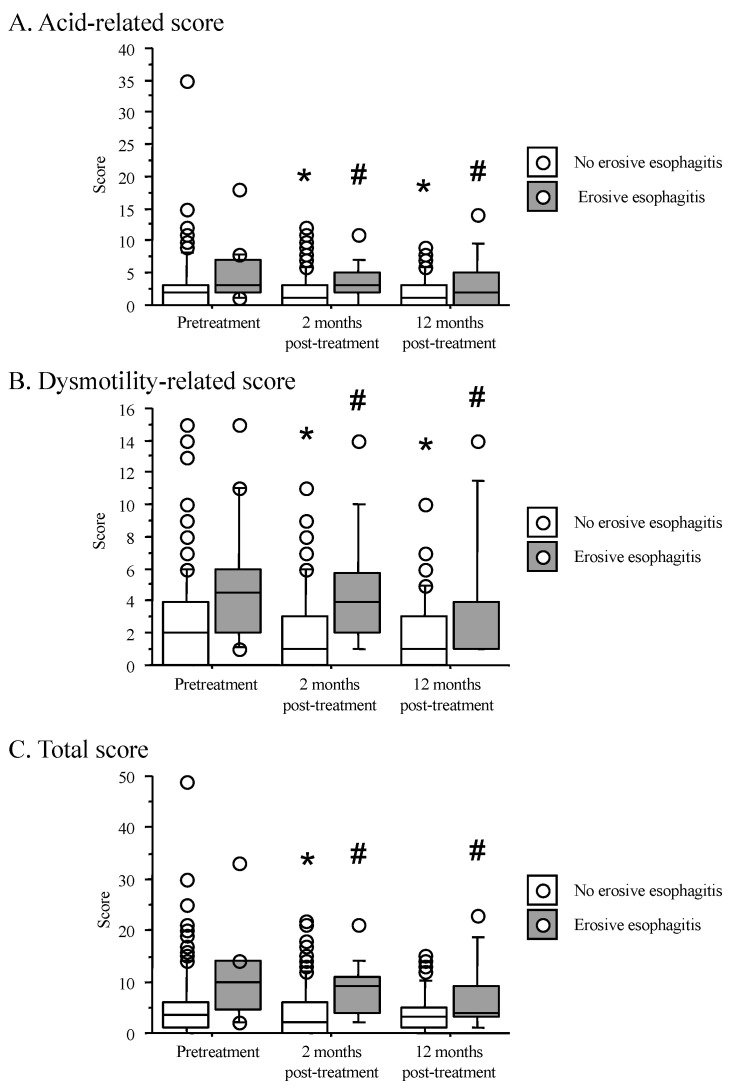
Correlation between F-scale questionnaire score and time course from baseline to 2 months after treatment and 12 months after eradication success between patients with reflux esophagitis and nonerosive esophagitis. Scores in the F-scale questionnaire are shown as acid-related (**A**), dysmotility-related (**B**), and total scores (**C**). *: *p* < 0.05 vs. score at pretreatment and #: *p* < 0.05 vs. score of patients with nonerosive esophagitis.

**Table 1 jcm-10-01434-t001:** Characteristics of patients positive for *Helicobacter pylori* at baseline.

	All Patients (*n* = 148)	Non-GERD (*n* = 113)	Redness (*n* = 32)	Reflux Esophagitis (*n* = 3)	*p* Value
Age (years)	65.5 ± 10.3	65.9 ± 10.5	64.1 ± 9.8	68.7 ± 5.0	0.50
Sex (male/female, *n*/*n*)	78/70	58/55	17/15	3/0	0.25
Height (cm)	161.8 ± 7.9	161.7 ± 8.1	162.0 ± 7.3	163.3 ± 6.1	0.92
Body weight (kg)	58.9 ± 10.9	59.3 ± 10.8	57.5 ± 11.6	57.7 ± 6.7	0.84
Smoking (no/previous/current, *n*/*n*/*n*)	85/50/13	71/34/8	14/14/4	0/2/1	0.07
Alcohol (no/previous/current)	71/13/64	57/13/43	14/0/18	0/0/3	0.05
Hiatal hernia (−/+)	128/20	98/15	27/5	3/0	0.74
SSBE (−/+)	106/42	85/28	19/13	2/1	0.21
GERD (-/redness/grade A/grade B)	113/32/2/1	113/0/0/0	0/32/0/0	0/0/2/1	<0.01
Endoscopic Kyoto classification					
Atrophy	2 (0–2)	2 (0–2)	2 (1–2)	2 (2–2)	0.93
Intestinal metaplasia	1 (0–2)	1 (0–2)	1 (0–2)	1 (0–2)	0.79
Diffuse redness	2 (0–2)	2 (0–2)	2 (1–2)	1 (1–2)	0.16
Total score	5 (2–7)	5 (2–7)	5 (2–7)	4 (4–7)	0.26
Eradication history (1st/2nd/3rd)	107/26/15	84/19/10	22/7/3	1/0/2	0.02
F scale					
Acid-related score	2 (0–35)	2 (0–35)	2 (0–11)	7 (0–8)	0.37
Dysmotility-related score	2 (0–15)	2 (0–15)	2 (0–10)	4 (0–6)	0.61
Total score	4 (0–49)	4 (0–49)	4 (0–19)	11 (1–14)	0.51

BMI, body mass index; GERD, gastroesophageal reflux diseases; SSBE, short-segment Barrett’s esophagus. Values for age, height, and body weight are shown as the mean ± standard deviation. Scores for the Endoscopic Kyoto classification and questionnaire are shown as median (range).

**Table 2 jcm-10-01434-t002:** Characteristics of patients at 12 months after eradication success.

	All Patients (*n* = 148)	Non-GERD (*n* = 106)	Redness (*n* = 26)	Reflux Esophagitis (*n* = 16)	*p* Value
GERD (-/redness/grade A/grade B)	106/26/10/6	106/0/0/0	0/26/0/0	0/0/10/6	
Hiatal hernia (−/+)	106/42	77/29	22/4	12/4	0.31
SSBE (−/+)	128/20	94/12	19/7	10/6	0.69
Endoscopic Kyoto classification					
Atrophy	2 (0–2)	2 (1–2)	2 (0–2)	2 (1–2)	0.80
Intestinal metaplasia	1 (0–2)	1 (0–2)	1 (0–2)	1 (0–2)	0.59
Diffuse redness	2 (0–2)	2 (0–2)	2 (1–2)	2 (1–2)	0.93
Total score	5 (2–7)	5 (2–7)	4.5 (2–7)	5 (2–7)	0.68
Questionnaire					
F scale					
Acid-related score	3.5 (0–14)	3 (0–8)	3.5 (0–9)	5 (0–14)	0.14
Dysmotility-related score	1 (0–14)	1 (0–10)	2 (0–7)	1 (0–14)	0.16
Total score	3 (0–23)	3 (0–14)	7.5 (0–15)	6 (1–23)	0.05

GERD, gastroesophageal reflux diseases; SSBE, short-segment Barrett’s esophagus. Scores for the Endoscopic Kyoto classification and questionnaire are shown as median (range).

**Table 3 jcm-10-01434-t003:** Association of severity of GERD with pretreatment and after treatment findings.

Post-Treatment	Non-GERD (*n* = 106)	Redness (*n* = 26)	GERD Grade A (*n* = 10)	GERD Grade B (*n* = 6)
Pretreatment				
Non-GERD (*n* = 113)	91	14	5	3
Redness (*n* = 32)	14	12	5	1
GERD grade A (*n* = 1)	0	0	0	1
GERD grade B (*n* = 2)	1	0	0	1

GERD, gastroesophageal reflux diseases.

**Table 4 jcm-10-01434-t004:** Association of category with reflux esophagitis and reflux-related symptoms before and after eradication therapy.

Category			Before Eradication Therapy					After eradication Therapy, 12 Months				
Sex	SSBE	Hiatal Hernia	Number	Non-GERD/ GERD (*n*/*n*)	F Scale Acid-Related Score	F Scale Dysmotility-Related Score	F Scale Total Score	Number	Non-GERD/GERD (*n*/*n*)	F Scale Acid-Related Score	F Scale Dysmotility-Related Score	F Scale Total Score
Male	−	−	47	36/11	2 (0–10)	1 (0–7)	4 (0–17)	47	32/15	0 (0–9)	1 (0–14)	2 (0–15)
Male	−	+	5	5/0	0 (0–3)	1 (0–3)	1 (0–5)	5	5/0	0 (0–0)	1 (1–1)	1 (1–1)
Male	+	−	23	16/7	1 (0–18)	2 (0–15)	3 (0–33)	23	16/7	1 (0–5)	1 (0–4)	2 (0–9)
Male	+	+	3	1/2	8 (3–12)	6 (2–13)	14 (5–25)	3	2/1			
Female	−	−	46	39/7	1 (0–15)	2 (0–15)	4 (0–30)	46	37/9	1.5 (0–7)	2 (0–10)	3 (0–14)
Female	−	+	8	5/3	4 (1–35)	3 (0–14)	6.5 (1–49)	8	3/5	4 (3–6)	3 (0–5)	9 (3–9)
Female	+	−	12	7/5	2 (0–9)	3.5 (0–10)	5.5 (0–19)	12	9/3	3 (0–5)	2 (0–7)	5 (0–12)
Female	+	+	4	4/0	6.5 (4–7)	6 (2–7)	13 (6–13)	4	2/2	9.5 (5–14)	7 (5–9)	16.5 (10–23)
*p* Value					<0.01	0.12	0.03			-	-	-

GERD, gastroesophageal reflux diseases; SSBE, short-segment Barrett’s esophagus. Scores for the questionnaire are shown as median (range).

**Table 5 jcm-10-01434-t005:** Univariate and multivariate analysis for the development of reflux esophagitis.

	Univariate Analysis	Multivariate Analysis
Parameters	Odds Ratio	95% CI	*p* Value	Odds Ratio	95% CI	*p* Value
Age (years)	0.985	0.939–1.034	0.540	0.975	0.925–1.028	0.356
Sex (male, vs. female)	2.134	0.703–6.482	0.181	3.499	0.987–12.399	0.052
Hiatal hernia	2.417	0.695–8.406	0.165	2.312	0.533–10.067	0.261
short segment Barrett’s esophagus	1.600	0.542–4.722	0.395			
Smoking	2.394	0.429–13.373	0.320			
Alcohol	1.496	0.523–4.282	0.453			
Kimura–Takemoto (moderate)	0.398	0.064–2.464	0.322			
Kimura–Takemoto (severe)	0.389	0.070–2.163	0.389			
Endoscopic Kyoto classification						
Atrophy	1.094	0.248–4.833	0.905			
Intestinal metaplasia	1.443	0.711–2.929	0.310			
Diffuse redness	1.075	0.388–2.977	0.890			
Total score	1.083	0.702–1.672	0.718			
F scale, pretreatment						
Acid-related score	1.085	0.989–1.189	0.084			
Dysmotility-related score	1.200	1.055–1.364	0.005			
Total score	1.069	1.009–1.132	0.023	1.069	1.003–1.139	0.039
F scale, 2 months after treatment	1.130	1.037–1.233	0.006

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
