# Peer review of "Risk of Reflux-Related Symptoms and Reflux Esophagitis after Helicobacter pylori Eradication Treatment in the Japanese Population"

_jcm, 2021, doi:10.3390/jcm10071434_

Round 1
Reviewer 1 Report
The paper was extensively rewritten and recalculated resulting in a clearer presentation and conclusion.
Minor:
page 6 figure 1: in the legend to figure 1 # is given vor p < 0.05. However in the figure itself no such sign is shown. Probably because vs score at 2 months was not significant. Please check
page 7 lines 233 ff: F scale scores ....were higher....: was this significant? for all scores? pleas check and if so provide a p value. You might consider omitting table 4 and just report a little more extensive in the text.
page 9 line 273: should probably read acid reflux not reflex.
Author Response
Our responses to comments raised by the Reviewer 1
- page 6 figure 1: in the legend to figure 1 # is given vor p < 0.05. However, in the figure itself no such sign is shown. Probably because vs score at 2 months was not significant. Please check.
Response:
As your comments, “#” was not used in the Figure 1 of this present version. The “#” was used at the original version of the first submission and Figures used “#” was deleted by Reviewer’s recommendation. Therefore, we deleted this part in the revised version.
- page 7 lines 233 ff: F scale scores ....were higher....: was this significant? for all scores? pleas check and if so provide a p value. You might consider omitting table 4 and just report a little more extensive in the text.
Response:
Thank you for your comments. Because this table was recommended to add in this paper by another reviewer, it might be difficult to omit this table.
Therefore, we added p values in Table 4 of revised version. Unfortunately, although it is not able to show p values after eradication therapy, because of loss of data in patients of male/SSBE(+)/hernia(+), the acid related and total F scale scores at before eradication therapy significantly differed among different groups based on sex, SSBE and hiatus hernia. The F scale scores in patients with both SSBE and hiatus hernia were higher than SSBE-negative, hiatal hernia-negative or both-negative patients, irrespective of sex, before eradication therapy.
- page 9 line 273: should probably read acid reflux not reflex.
Response:
This is our mistake. In the revised version, we changed “reflex” in “reflux”.
Reviewer 2 Report
Sample size is ok but not great. Also study is restricted geographically (which in this particular diagnosis/association has great importance)
Design is adequate although could be improved as few confounding factors have not been taken into account ( like location of H. pylori infection)
Statistics are adequate. English is adequate.
Although, my main concern if about the hypothesis that is being used to generate the study .
H. pylori and gastric acid production is a complex association with decades of research and many pathways and theories have been established. In summary, the effect of H. pylori on gastric acid production depends on the location and chronicity of HP in the stomach. It can lead to increased acid production (when antrum predominant) with consequent duodenal ulcers versus decreased acid production (direct inhibitory effect on oxyntic mucosa) leading to atrophic gastritis and consequent risk of gastric adenocarcinoma.
Also, research is well established that reflux is a result of transient relaxation of lower esophageal sphincter and is actually independent of gastric acid production. Also anti-acid treatment for GERD improves symptoms by decreasing the acid component in reflux but the number of reflux episodes stay same (pt just do not feel it due to non acid content mostly).
So, in my opinion, these two (reflux and H. pylori) are two completely different diseases with no clear proven physiologic pathways to effect each other.
Overall, the current study is unlikely to add clinically useful information to current existing literature.
Author Response
Our responses to comments raised by the Reviewer 2
- Sample size is ok but not great. Also study is restricted geographically (which in this particular diagnosis/association has great importance)
Response:
As your comments, sample size of this study is not great. In addition, this study was performed in Japan, as a single-center retrospective study, suggested that this study might have bias of diagnosis/association for esophagitis and be limitation. Therefore, we added these comments as limitation in the revised version.
- Design is adequate although could be improved as few confounding factors have not been taken into account (like location of pylori infection)
Response:
We agree with your comments and also think that It will be better to taken into account for confounding factors. However, because ample size of this study is not great, we will plan multicenter prospective study as further study by considering confounding factors in feature.
We added comments to plan multicenter prospective study by considering confounding factors in the revised version.
- Statistics are adequate. English is adequate.
Response:
Thank you for your comments.
- Although, my main concern if about the hypothesis that is being used to generate the study.
- pylori and gastric acid production is a complex association with decades of research and many pathways and theories have been established. In summary, the effect of H. pylori on gastric acid production depends on the location and chronicity of HP in the stomach. It can lead to increased acid production (when antrum predominant) with consequent duodenal ulcers versus decreased acid production (direct inhibitory effect on oxyntic mucosa) leading to atrophic gastritis and consequent risk of gastric adenocarcinoma.
Also, research is well established that reflux is a result of transient relaxation of lower esophageal sphincter and is actually independent of gastric acid production. Also anti-acid treatment for GERD improves symptoms by decreasing the acid component in reflux but the number of reflux episodes stay same (pt just do not feel it due to non acid content mostly).
So, in my opinion, these two (reflux and H. pylori) are two completely different diseases with no clear proven physiologic pathways to effect each other.
Response:
Thank you for your comments and suggestion.
As your comments, gastric acidity differs between patients with antrum predominant gastritis and patients with severe gastric mucosal atrophy. When pyloric gastritis is predominant, IL-8 primarily stimulates gastrin-producing cells, resulting in hypergastrinemia and a consequent increase in acid secretion. Patients with pyloric-predominant gastritis are therefore at higher risk of duodenal ulcer and likely, also reflux esophagitis. The effects of IL-1beta is mainly observed after the extension of severe atrophy. IL-1beta inhibits acid secretion with 100-fold greater potency than PPIs. Therefore, when body gastritis becomes dominant, acid secretion is substantially suppressed. When this stage is reached, patients are at an increased risk of gastric ulcers and cancer, while the risk of esophagitis decreases.
Eradication leads to the resolution of inflammation in the gastric fundic mucosa. The recovery of acid secretion that follows this resolution has led to concerns about the development of esophagitis. In such patients with pyloric-predominant gastritis and potent acid secretion, eradication reduces gastrin stimulation by IL-8 and normalizes acid secretion, which is expected to prevent esophagitis. The findings indicate that the degree of gastritis at the time of eradication influences the recovery of acid secretion and the subsequent risk of esophagitis after eradication. Therefore, we think that the incidence rate of esophagitis should be distinguished between both groups.
In general, the etiology of GERD is multifactorial and includes frequent and prolonged reflux of gastric contents, status and size of hiatal hernia, dysfunction of LES, dysfunction of esophageal motility, and hypersensitivity. Both the increase in acid secretion and decrease in LES pressure play important roles in the development of esophagitis. Therefore, we think that gastric acid is one of necessary conditions for esophagitis, but not sufficient conditions. Reviewer suggested that reflux esophagitis and H. pylori infection are two completely different diseases. We agree your opinion. However, because H. pylori infection is inversely associated with esophagitis in systematic reviews and meta-analysis and that we often experience patients with esophagitis and reflux-related symptoms after eradication treatment, we believe that although it may not be the main cause, H. pylori infection and severity of gastric mucosal atrophy influence development of esophagitis.
Reviewer 3 Report
- Patients - the authors need to mention what the standard of care is in Japan in-so-far as followup following treatment of H pylori. Is it usual for all patients to undergo a second endoscopy after 12 months and have 2 H. pylori eradications performed?
- The patients were diagnosed as being HP positive if 1 of 3 tests were positive. IgG for HP is not considered a marker of current infection, rather of past exposure to the bacterium. Even though none of the patients ended up having "only" positive IgG - this test should not be recommended as a test for current infection.
- When did you start counting the time - from the end of the last treatment protocol leading to successful eradication or from the first eradication attempt? Figure 2 would indicated that the clock started after the first eradication - which means that patients with failed eradication received more acid suppression during the 12 months than did those successfully eradicated.
- The differences in rates of successful eradication also influence the time under acid suppression prior to successful eradication, which consequently may have influenced the results concerning GERD signs and symptoms.
- What was the extent of PPI/H2blocker exposure prior to eradication attempts (for dypeptic symtpoms, epigastric pain , etc...? This might also affect the low rate of symptoms prior to eradication, furthermore, patients with HP gastritis, may have abdominal complaints that "take precedence" over the reflux symptoms - and therefore downplay the reflux symptoms prior to eradication.
- Line 135 I think you mean EG not EC
- Line 137 - I think you mean "findings were confirmed histologically".
- Did patients with known GERD/reddness have any additional acid suppression treatment except for their 7day HP treatment? (either from you or from family doctors for symptoms - since this was not a prospective study, they would not have been told not to take acid suppression during the year for symptoms, and standard of care would demand that a patient with active GERD be treated. This is mentioned briefly in the limitations - but is a Major methodological problem with this study.
Author Response
Our responses to comments raised by the Reviewer 2
- Patients - the authors need to mention what the standard of care is in Japan in-so-far as followup following treatment of H pylori. Is it usual for all patients to undergo a second endoscopy after 12 months and have 2 H. pylori eradications performed?
Response
Thank you for your comments.
In Japan, as the standard of care, H. pylori eradication therapies are limited by Japanese insurance system to regimens comprising an acid-inhibitory drug such as a PPI or VPZ at a standard dose bid, amoxicillin 750 mg bid, and clarithromycin 200 mg or 400 mg bid for 7 days as a first-line eradication regimen; and PPI or VPZ bid, amoxicillin 750 mg bid, and metronidazole 250 mg bid for 7 days as a second-line eradication regimen, irrespective of infection of resistance strains to antimicrobial agents.
It is usually recommended to undergo a second endoscopy after 12 months for all patients by the Guidelines for the management of Helicobacter pylori infection in Japan. Patients are usually performed H. pylori eradications one time. If eradications are failed, second-line therapy is recommended to perform.
According to your comments, we mention the standard care of eradication regimens and follow-up in the revised version in the revised version.
- The patients were diagnosed as being HP positive if 1 of 3 tests were positive. IgG for HP is not considered a marker of current infection, rather of past exposure to the bacterium. Even though none of the patients ended up having "only" positive IgG - this test should not be recommended as a test for current infection.
Response
We agree your comments. In the revised version, we deleted sentences of H. pylori IgG test to avoid misleading, as below.
- pylori infection was diagnosed in all patients using the rapid urease test (Helicocheck®; Institute of Immunology, Co., Ltd., Tochigi, Japan) and a culture test (BML, Inc., Tokyo, Japan). Patients were diagnosed as H. pylori infection-positive if at least one of the two tests was positive.
- When did you start counting the time - from the end of the last treatment protocol leading to successful eradication or from the first eradication attempt? Figure 2 would indicated that the clock started after the first eradication - which means that patients with failed eradication received more acid suppression during the 12 months than did those successfully eradicated.
Response
Thank you for your comments.
As below, endoscopy was performed at entry and 12 months after evaluation of eradication success by a 13C-urea breath test. The F scale questionnaire was performed at entry, 2 months post-treatment, 12 months after evaluation of eradication success. Therefore, total observational period in patients with initial successful eradication was 14 months and that in patients with failed eradication was 16 months.
In the revised version, we revised time schedule of endoscopy and F scale questionnaire to avoid misleading.
- The differences in rates of successful eradication also influence the time under acid suppression prior to successful eradication, which consequently may have influenced the results concerning GERD signs and symptoms.
Response
Previous meta-analysis which reported by Janssen MJ1 showed that pre-treatment with a PPI does not influence H. pylori eradication outcome. However, as suggested by you, there is possibility that time under acid suppression prior to successful eradication may influence eradication outcome. In this study, we did not recruit patients receiving vonoprazan, PPI or H2RA. Therefore, we added this fact in the Material methods section of the revised version, as below.
Inclusion criteria were age ≥ 20 years with H. pylori infection, no medication by vonoprazan, proton pump inhibitor (PPI) and histamine 2 receptor antagonist (H2RA), performance of endoscopy to evaluate reflux esophagitis before eradication therapy and at 12 months post-eradication success and use of questionnaire to evaluate reflux-related symptoms before eradication therapy and at 2 months post-treatment and 12 months post-eradication success in our University Hospital, and conclusive evaluation of eradication outcome by the 13C-urea breath test (UBIT 100 mg tablets, Otsuka Pharmaceutical Co., Ltd., using a cut-off of 2.5‰).
Reference
- Janssen MJ, et al. Meta-analysis: the influence of pre-treatment with a proton pump inhibitor on Helicobacter pylori eradication. Aliment Pharmacol Ther. 2005; 21:341-5
- What was the extent of PPI/H2blocker exposure prior to eradication attempts (for dypeptic symtpoms, epigastric pain , etc...? This might also affect the low rate of symptoms prior to eradication, furthermore, patients with HP gastritis, may have abdominal complaints that "take precedence" over the reflux symptoms - and therefore downplay the reflux symptoms prior to eradication.
Response
We agree your comments. Medication of PPI/H2RA generally related with the rate of reflux esophagitis and reflux-relatedsymptoms prior to eradication. In addition, patients with H. pylori-associated atrophic gastritis, may have abdominal complaints that "take precedence" over the reflux-related abdominal symptoms. We excluded patients taking vonoprazan, PPI or H2RA at the entry to avoid these effects in this study.
We added this fact in the Material methods section of the revised version, as above (Please see answer for question 4).
- Line 135 I think you mean EG not EC
Response
Thank you for your comments. This was our mistake. We revised “EC” in “EG” in the revised version.
- Line 137 - I think you mean "findings were confirmed histologically".
Response
In this study, we did not perform histopathological evaluation. We did diagnosis of GERD by endoscopy alone. We deleted this sentence, “These findings were confirmed by endoscopy“, to avoid misunderstandings in the revised version.
- Did patients with known GERD/reddness have any additional acid suppression treatment except for their 7day HP treatment? (either from you or from family doctors for symptoms - since this was not a prospective study, they would not have been told not to take acid suppression during the year for symptoms, and standard of care would demand that a patient with active GERD be treated. This is mentioned briefly in the limitations - but is a Major methodological problem with this study.
Response
Thank you for your comments.
After successful H. pylori eradication therapy, two patients were received additional PPI therapy for the reflux-related abdominal symptoms in this study. We agree your comments and this is the limitations as the Major methodological problem with this study. Therefore, we added this fact as limitation in the revised version, as below.
Forth, because we did not limit the medication of acid secretion inhibitors after H. pylori eradication therapy, two patients were received additional PPI therapy for the reflux-related abdominal symptoms after successful H. pylori eradication therapy. This is considered to be a major methodological problem.
Reviewer 4 Report
The manuscript by Mitsuhige Sugimoto et al. describe very interesting data on the implication of Hp eradication treatment on the reflux-related symptoms.
Some revision is needed before possible acceptance of the manuscript.
How was determined the number of patient to include? Please add to the manuscript .
The description of the method used for Hp culture and biological diagnosis is insufficient. Moreover, on this type of manuscript it could be useful to give the opportunity to a microbiologist to share his/her expertise. Please consider.
How was consider AST-orientated treatment in Japan? If possible please compare these empiric and orientated treatment or discuss it if not possible.
How was determined failure of eradication treatment ?
Could the pre/post-treatment finding been statistically associated (if clinically pertinent)?
For Figure 1 : please suppress "#! p<0.05 vs score at 2 months post treatment as this symbol could not beobserved in this figure.
Global : please italicized "vs." e.g." " et al." and bacterial names/genes.
Author Response
Our responses to comments raised by the Reviewer 3
- How was determined the number of patient to include? Please add to the manuscript.
Response
Thank you for your comments.
We determined the number of patients in this study, as below. With agreement to your comments, we added how to determine the number of patients in the revised version.
The sample size and power were calculated by the t-test at set that the effects size is 0.3 and the correlation coefficient is 0.6, desired power 80%, with a significance level of 0.05 in a two-sided test and the required sample number was 134. In addition, we expected that 10% of patients enrolled in the study would delete by loss of data; we therefore aimed to entry 148 patients for a valid analysis.
- The description of the method used for Hp culture and biological diagnosis is insufficient. Moreover, on this type of manuscript it could be useful to give the opportunity to a microbiologist to share his/her expertise. Please consider.
Response
Thank you for your comments.
We added any description how to culture H. pylori in the revised version, as below.
For bacterial culture and antimicrobial sensitivity testing, agar plates were inoculated with biopsy specimens and incubated at 37 °C under microaerophilic conditions (5% O2, 10% CO2 and 85% N2) for approximately 7 days at 37 ℃. H. pylori was identified using oxidase production. H. pylori colonies were subcultured using the agar dilution method to determine the minimum inhibitory concentration (MIC) for amoxicillin, metronidazole, clarithromycin and sitafloxacin, according to the recommendations of the Clinical and Laboratory Standards Institute (CLSI) and the manufacturer’s instructions. Cut-off MICs used to define resistance were > 1.0 µg/mL for clarithromycin and sitafloxacin and > 8 µg/mL for metronidazole. For amoxicillin, the cut-off MICs used to define resistance and the absence of sensitivity were > 0.5 µg/mL and > 0.06 µg/mL, respectively.
- How was consider AST-orientated treatment in Japan? If possible please compare these empiric and orientated treatment or discuss it if not possible.
Response
Thank you for your comments.
Now, the AST-orientated treatment in Japan is major for most of infectious disease.
In Japan, however, H. pylori eradication therapies are currently limited by Japanese insurance system to regimens comprising an acid-inhibitory drug such as a PPI or VPZ at a standard dose bid, amoxicillin 750 mg bid, and clarithromycin 200 mg or 400 mg bid for 7 days as a first-line eradication regimen; and PPI or VPZ bid, amoxicillin 750 mg bid, and metronidazole 250 mg bid for 7 days as a second-line eradication regimen, irrespective of infection of resistance strains to antimicrobial agents. Therefore, the AST-orientated treatment is not given for H. pylori infection in Japan. I think that susceptibility-based tailored treatment should be selected for H. pylori eradication therapies in Japan. We added any comments in Discussion of the revised version
- How was determined failure of eradication treatment?
Response
In this study, eradication success was evaluated using a 13C-urea breath test with a cut-off value of 2.5 ‰ at 6–8 weeks after treatment. We added how to determine eradication outcome in the revised version, as “Eradication success was evaluated using a 13C-urea breath test with a cut-off value of 2.5 ‰ at 6–8 weeks after treatment”.
- Could the pre/post-treatment finding been statistically associated (if clinically pertinent)?
Response
In this study, because total baseline F scale score was again shown to be a significant risk factor for development of reflux esophagitis (OR: 1.069, 95% CI: 1.003–1.139, p = 0.04) on multivariate analysis using factors showing a p value <0.2 in the univariate analysis (hiatal hernia and total score of F scale) and age and sex as adjustment factors, and the total score for the F scale questionnaire at 2 months post-treatment showed significant differences between patients with successful and failed treatment (p = 0.04), we believe that the pre/post-treatment finding is statistically associated.
- For Figure 1: please suppress "#! p<0.05 vs score at 2 months post treatment as this symbol could not beobserved in this figure.
Response
As your comments, “#” was not used in the Figure 1 of this present version. The “#” was used at the original version of the first submission and Figures used “#” was deleted by Reviewer’s recommendation. Therefore, we deleted this part in the revised version.
- Global: please italicized "vs." e.g." " et al." and bacterial names/genes.
Response
Thank you for your comments. We carefully checked throughout manuscript and italicized "vs." “e.g." " et al." and bacterial names/genes in the revised version.
Round 2
Reviewer 2 Report
Authors have answered all the query that were raised on prior review.
I have no other doubts about the study.
Author Response
Our responses to comments raised by the Reviewer 2
- Authors have answered all the query that were raised on prior review.
I have no other doubts about the study.
Response:
Thank you for your comments. We believe that this study will be of a great interest to the readers of your Journal.
Reviewer 3 Report
The authors have made changes to the manuscript to improve the scientific integrity.
They wrote that the followup was 12-14 months, depending on if the patient passed or failed 1st line therapy. What about if patients failed 2nd line treatment?
Author Response
Our responses to comments raised by the Reviewer 3
- The authors have made changes to the manuscript to improve the scientific integrity.
They wrote that the followup was 12-14 months, depending on if the patient passed or failed 1st line therapy. What about if patients failed 2nd line treatment?
Response
Thank you for your comments.
As the initial eradication therapy in our hospital, all patients underwent eradication with vonoprazan and a combination of two antibiotics, namely clarithromycin and amoxicillin as first-line eradication treatment (n = 107), metronidazole and amoxicillin as second-line treatment (n = 26), and sitafloxacin and amoxicillin as third-line treatment (n = 15). When initial eradication therapy failed in our hospital (n = 20), patients received advanced eradication therapy, namely a second-line eradication regimen for patients who failed a first-line regimen, a third-line regimen for patients who failed a second-line regimen and a fourth-line regimen [sitafloxacin and metronidazole] for those who failed a third-line regimen. Twenty patients who failed initial eradication therapy received advanced eradication therapy, was successful in all patients.
We added this in the revised version.
Reviewer 4 Report
After extensive revision, I think that now, the manuscript is suitable for publication.
Author Response
Our responses to comments raised by the Reviewer 4
- After extensive revision, I think that now, the manuscript is suitable for publication.
Response
Thank you for your comments. We believe that this study will be of a great interest to the readers of your Journal.
This manuscript is a resubmission of an earlier submission. The following is a list of the peer review reports and author responses from that submission.
Round 1
Reviewer 1 Report
Ths manuscript by Mitsushige Sugimoto et al. describe an interesting study on the esophagitis symptomatology before and after Hp eradication.
This article deserve some revision before possible acceptance.
General :
- "i.e." deserve to be written in italic.
Manuscript :
- Details have to be given on the rapid urease test, the serology and the culturing of Helicobacter. Adding some microbiologists to the authors' list could be of interest for them to bring their expertise and point of view.
- How many patient receive the "advanced eradication therapy"? Could you give more detail (molecule, regimen, results)? Have this category been considered considered in the results?
- In general it could be of interest to stratify results according to the number of lines, as the symptomatology could be different between these groups. please complete.
- Table 1 : For bimodal characteristics (as sex, hiatal hernia, SSBE, ...)
Author Response
Our responses to comments raised by the Reviewer 1
- "i.e." deserve to be written in italic.
Response:
According to your comment, we revised "i.e." in "i.e." in the revised version.
- Details have to be given on the rapid urease test, the serology and the culturing of Helicobacter. Adding some microbiologists to the authors' list could be of interest for them to bring their expertise and point of view.
Response:
Thank you for you comments. In this study, because we firstly focused on association with H. pylori eradication and development of reflux esophagitis/reflux-related symptoms, we did not mention about detail information about status of H. pylori infection, detection methods and measurement of antibiotic resistance in the original version. However, according to your comment, we added detail information in the revised version, as below:
- pylori infection was diagnosed in all patients using the rapid urease test, an anti-H. pylori IgG test (>10), and a culture test. Patients were diagnosed as H. pylori infection-positive if at least one of the three tests was positive. For bacterial culture and antimicrobial sensitivity testing, agar plates were inoculated with biopsy specimens and incubated at 37 °C under microaerophilic conditions. H. pylori colonies were subcultured using the agar dilution method to determine the minimum inhibitory concentration (MIC) for amoxicillin, metronidazole, clarithromycin and sitafloxacin. Cut-off MICs used to define resistance were > 1.0 µg/mL for clarithromycin and sitafloxacin and > 8 µg/mL for metronidazole. The cut-off MICs used to define resistance and the absence of sensitivity were > 0.5 µg/mL and > 0.06 µg/mL for amoxicillin, respectively.
In this study, 83.4% (207/242) patients for IgG, 80.4% (193/240) for culture and 92.1% (258/280) patients for rapid urease test were positive for H. pylori infection. However, there was no patients of H. pylori-IgG (positive), rapid urease test (negative) and culture test (negative).
- How many patients receive the "advanced eradication therapy"? Could you give more detail (molecule, regimen, results)? Have this category been considered in the results?
Response:
In this study, when initial eradication therapy failed, patients received advanced eradication therapy. Eradication rates in the ITT analysis were 86.4% (191/220) for first-line therapy, 91.2% (44/48) for second-line therapy, and 95.2% (40/42) for third-line therapy. Therefore, of the 310 H. pylori-positive Japanese patients, 35 patients who failed initial eradication therapy received advanced eradication therapy, which was successful in all patients. Therefore, in this study, all H. pylori-positive Japanese patients were eradicated by any of regimen, we investigated association with H. pylori eradication and reflux esophagitis/ reflux-related symptoms.
According to your comment, we added detail information in the revised version, as below:
Method: Line 123-126
When initial eradication therapy failed, patients received advanced eradication therapy: a second-line eradication regimen for patients who failed a first-line regimen, a third-line regimen for patients who failed a second-line eradication regimen and a forth-line regimen [(sitafloxacin (100 mg bid) and metronidazole (250 mg bid)) for patients who failed a third-line eradication regimen.
Result: Line 170-174
Eradication rates in the ITT analysis were 86.4% (95% CI: 81.6%–91.0%, 191/220) for first-line therapy, 91.2% (95% CI: 80.0%–97.7%, 44/48) for second-line therapy, and 95.2% (95% CI: 83.4%–99.4%, 40/42) for third-line therapy. Thirty-five patients who failed initial eradication therapy received advanced eradication therapy, which was successful in all patients.
- In general, it could be of interest to stratify results according to the number of lines, as the symptomatology could be different between these groups. please complete.
Table 1 : For bimodal characteristics (as sex, hiatal hernia, SSBE, ...)
Response:
Thank you for your comments. As your comments, the number and kinds of reflux-related symptoms and the kinds of endoscopic findings may differ among different number of lines. Therefore, we add new Table, as Table 4 (Association with category related with reflux esophagitis and reflux-related symptoms), in the revised version, as below.
We investigated with association with reflux-related symptoms for the F scale questionnaire and category using factors (i.e., sex, SSBE and hiatal hernia) related with reflux esophagitis. The scores for the F scale questionnaire in patients with SSBE and hiatus hernia were higher than those without SSBE and/or hiatal hernia, irrespective with sex, in both before eradication therapy and 12 months post-treatment.
Reviewer 2 Report
The author´s aim to determine the risk of reflux related symptoms and reflux esophagitis in japanese patients after H. pylori eradication. (The abbreviation RR in the abstract is very misleading and should not be used)
They present an overwhelming amount of data that makes the paper difficult to read. Nevertheless this study has severe limitations, that make the results hard to interpret.
Less then half of the patients underwent follow up endoscopy at the recruiting center. Thus there is a strong possibility of bias in follow up.
The diagnosis of H. pylori infection was not done by biopsy, a gold standard in endoscopy based studies one would expect. In contrast IgG antibodies were accepted as a positive result, even when two other, more accurate tests, were negative. Thus it can be expected, that H. pylori negative patients are included. The results on basis of what test result the patients were included should be presented.
After 2 months the patients were re-evaluated with 3 different questionnaires. At this timepoint around 14% were not successfully eradicated. If the authors want to compare 3 different questionnaires, they should do so in a separate study. Otherwise the results section is much too long with 17 diagrams. In addition the Izumo scale is not very well known to readers outside of Japan.
What eradication therapy is recommended for patients who are allergic to penicilline?
Endoscopic assessment of reflux esophagitis: Reference 27 has nothing to do with the LA classification. It is a review of the first author including probably at least partially results of the present study. The NERD classification for reflux assessment should be better explained and referenced as it is not widely used outside Japan. It´s usage in the current study limits the comparability with western studies.
The authors have no explanation why acid related scores decreased after eradication but endoscopic reflux increased. The significant increase in the endoscopic findings might well due to the increased incidence of redness (= M NERD), a less well established finding as mentioned above.
The authors might aim to retrieve the over 160 endoscopy reports from the other centers after 12 month to strengthen or clarify their findings.
Author Response
Our responses to comments raised by the Reviewer 2
- The abbreviation RR in the abstract is very misleading and should not be used
Response
According for your comments, we avoided use of “RR” and used “reflux esophagitis” in Abstract of the revised version.
- They present an overwhelming amount of data that makes the paper difficult to read. Nevertheless this study has severe limitations, that make the results hard to interpret. Less then half of the patients underwent follow up endoscopy at the recruiting center. Thus there is a strong possibility of bias in follow up.
Response
Thank you for your comments. As your comments, because 162 patients said they wanted to undergo endoscopic evaluation of gastric condition after eradication therapy the following year at a nearby hospital or health check-up center on enquiry at that time of eradication, not all patients could be evaluated for the development of reflux esophagitis and reflux-related symptom at 12 months post-treatment. Therefore, there is a strong possibility of bias in follow up, as mentioned in limitation section of manuscript.
If you will require, we re-analysis association with eradication and reflux esophagitis using 148 patients with successful H. pylori eradication underwent endoscopy in our hospital at 12 months post-treatment, not all 310 patients received H. pylorieradication therapy in our hospital.
- The diagnosis of pylori infection was not done by biopsy, a gold standard in endoscopy based studies one would expect. In contrast IgG antibodies were accepted as a positive result, even when two other, more accurate tests, were negative. Thus it can be expected, that H. pylori negative patients are included. The results on basis of what test result the patients were included should be presented.
Response
As you suggested, IgG antibodies might be accepted as a positive result, even when culture and rapid urease test were negative. In this study, 83.4% (207/242) patients for IgG, 80.4% (193/240) for culture and 92.1% (258/280) patients for rapid urease test were positive for H. pylori infection. However, there was no patients of H. pylori-IgG (positive), rapid urease test (negative) and culture test (negative). Therefore, to avoid misleading, we added this result in Method section of the revised version (L100-103).
- After 2 months the patients were re-evaluated with 3 different questionnaires. At this timepoint around 14% were not successfully eradicated. If the authors want to compare 3 different questionnaires, they should do so in a separate study. Otherwise the results section is much too long with 17 diagrams.
In addition the Izumo scale is not very well known to readers outside of Japan.
Response
Thank you for your comments. As you suggested, we deleted data of GSRS and Izumo scale and 6 diagrams in the revised version. We used the F-scale questionnaire alone and showed its data in the revised version.
- What eradication therapy is recommended for patients who are allergic to penicilline?
Response
In Japan, the eradication regimen of clarithromycin + metronidazole or sitafloxacin + metronidazole are recommended for patients who are allergic to penicillin. However, there was no patient who is allergic to penicillin in this study.
- Endoscopic assessment of reflux esophagitis: Reference 27 has nothing to do with the LA classification. It is a review of the first author including probably at least partially results of the present study. The NERD classification for reflux assessment should be better explained and referenced as it is not widely used outside Japan. It´s usage in the current study limits the comparability with western studies.
Response
Thank you for your comments. We agree with your comments. In this study, we focused on endoscopic esophageal change of endoscopic redness and reflux esophagitis and did not focus on NERD. We revised the section of “2.3. Endoscopy and severity of gastritis” to delete part about NERD, as below, and deleted reference 27 in the revised version, as below.
Reflux esophagitis was assessed according to the Los Angeles classification (grade A to D). In addition, redness was endoscopically defined as mucosal findings of redness, edema, or white granules in the EC-junction, irrespective of the presence of reflux-related symptoms. GERD was defined as reflux esophagitis and redness. These findings were confirmed by endoscopy.
- The authors have no explanation why acid related scores decreased after eradication but endoscopic reflux increased. The significant increase in the endoscopic findings might well due to the increased incidence of redness (= M NERD), a less well established finding as mentioned above.
Response
Thank you for your comments. This is important point to explain why acid related scores decreased after eradication. In this study, reflux-related symptoms (scores) in patients with reflux esophagitis showed no remarkable changes among pretreatment and 2- and 12-months post- treatment, suggested that scores of patients without reflux esophagitis after eradication therapy decreased. In addition, reflux esophagitis developed de novo in 7.1% (8/113) in the non-GERD group at pretreatment and population of this group was not so many.
The Kyoto Global Consensus Meeting recently defined H. pylori-associated dyspepsia as a condition in which abdominal symptoms disappeared or improved 6–12 months after eradication treatment. The improvement in abdominal symptoms after eradication in H. pylori-positive patients might be attributable to improved H. pylori-related and functional dyspepsia-related symptoms rather than to GERD-related symptoms, irrespective with recover of acid secretory ability after eradication therapy.
We should investigate association with eradication therapy and improvement of reflux-related symptoms using 24-hour intragastric pH monitoring as further study. We added any comments why acid related scores decreased after eradication but endoscopic reflux increased in the revised version.
- The authors might aim to retrieve the over 160 endoscopy reports from the other centers after 12 month to strengthen or clarify their findings.
Response
We agree with your comments. If possible, we think that it will be better to retrieve the over 160 endoscopy reports from the other centers at the 12 months posttreatment. However, it will be hard to contact with patients and to find out which hospital patients did the endoscopy at the 12 months posttreatment. In addition, we think that most of patients do not evaluate the severity of reflux-related symptoms by the F scale questionnaire. Finally, because such process was not reviewed and approved by the Institutional Review Board of Shiga University of Medical Science, we will be required to ask re-review of our modified protocol.
We added some comments as limitation in the revised version, as below.
Line 323-325
Although we think that to retrieve endoscopy report and symptom sheet of such 162 patients is required, contact with patients will be hard and such process was not reviewed and approved by our Review Board at the present.
Round 2
Reviewer 1 Report
The authors have considered most of my comments.
A unique one remains. The authors have to give details about the interpretation reference (EUCAST? CLSI) for AST and manufacturer's details for all assays.
Moreover, after consultation, I completely agree with the comments of the other reviewer. So, I recommend to reject the manuscript.
Author Response
Our responses to comments raised by the Reviewer 1
- A unique one remains. The authors have to give details about the interpretation reference (EUCAST? CLSI) for AST and manufacturer's details for all assays.
Response:
We showed manufacturer's details for assays including 13C-urea breath test, rapid urease test, anti-H. pylori IgG test and culture test in the revised version.
In this study, because the Japanese Society for Helicobacter Research do not recommend to use the EUCAST, we used generally selected cut-off MICs in Japan, as shown in references. In the revised version, we add three references about the interpretation reference 1-3.
- 13C-urea breath test (UBIT 100 mg tablets, Otsuka Pharmaceutical Co., Ltd., using a cut-off of 2.5‰)
- rapid urease test (Helicocheck®; Institute of Immunology, Co., Ltd., Tochigi, Japan)
- anti- pylori IgG test (antibody determination kit, E-Plate Eiken H. pyloriantibody, using a cut-off of 10 U/ml)
- culture test (BML, Inc., Tokyo, Japan).
- Kobayashi I, Murakami K, Kato M, et al. Changing antimicrobial susceptibility epidemiology of Helicobacter pylori strains in Japan between 2002 and 2005. J Clin Microbiol 2007;45:4006-10.
- Sugimoto M, Sahara S, Ichikawa H, et al. High Helicobacter pylori cure rate with sitafloxacin-based triple therapy. Aliment Pharmacol Ther 2015;42:477-83.
- Murakami K, Okimoto T, Kodama M, et al. Sitafloxacin activity against Helicobacter pylori isolates, including those with gyrA mutations. Antimicrob Agents Chemother 2009;53:3097-9.
Reviewer 2 Report
The authors provide a much improved version of their manuscript. However the main point of my criticism remains. More than half of the patients were lost to follow up, or to put it the other way around: of 310 patients screened, only 148 were included.
So yes, i suggest a recalculation of the 148 pat. included
minor points:
all red marked areas as provided by the authors should be corrected with a native speaker of the english language.
Especially line 100-101
line 135: GERD was defined as reflux and redness.... should it not read reflux esophagitis or redness?
lines 237-39: ??
lines 327-330: i do not understand what you mean: in line 320 you state that there was a significant decrease in the f score and here you state the opposite. The statement in line 320 is correct, so why do you argue against your results?
Author Response
Our responses to comments raised by the Reviewer 2
- The authors provide a much improved version of their manuscript. However, the main point of my criticism remains. More than half of the patients were lost to follow up, or to put it the other way around: of 310 patients screened, only 148 were included.
So yes, i suggest a recalculation of the 148 pat. Included
Response
Thank you for your comments. We agree with your recommendation.
In this revised version, we evaluated the prevalence of reflux esophagitis at 12 months post-treatment and the severity of reflux-related symptoms at 2 and 12 months post-treatment in 148 H. pylori-positive subjects who underwent H. pylori eradication.
Therefore, we revised Table 1, Figure 3 and Table 4 and contents.
- All red marked areas as provided by the authors should be corrected with a native speaker of the English language.
Response
Thank you for your comments. As your comments, this revised version was edited by Guy Harris DO from DMC Corp. (www.dmed.co.jp <http://www.dmed.co.jp/>), again.
Line 100-101→Line 103-105 in the revised version
In this study, 84.2% (101/120) of patients were positive by IgG, 80.9% (89/110) by culture and 91.9% (124/135) by the rapid urease test. No patient who was H. pylori-IgG -positive was negative for both the rapid urease test and culture.
line 135→Line 133-137 in the revised version
Reflux esophagitis was assessed according to the Los Angeles classification (grade A to D) [27]. In addition, redness was endoscopically defined as mucosal findings of
redness, edema, or white granules in the EC junction, irrespective of the presence of reflux-related symptoms. GERD was defined as reflux esophagitis or redness. These findings were confirmed by endoscopy.
lines 237-39→Line 233-237 in the revised version
We evaluated associations with scores of the F scale questionnaire and different categories using sex, SSBE and hiatal hernia (Figure 4). The F scale scores in patients with both SSBE and hiatus hernia were higher than SSBE-negative, hiatal hernia-negative or both-negative patients, irrespective of sex, both before eradication therapy and at 12 months post-treatment (Table 4).
- lines 327-330: i do not understand what you mean: in line 320 you state that there was a significant decrease in the f score and here you state the opposite.
Response
Thank you for comment. In this study, summary of acid-reflux scores is below:
|
Patients |
|
|
All patients (n = 148) |
decrease from pretreatment to post-treatment (Figure 1) |
|
Patients with esophagitis |
no change from pretreatment to post-treatment (Figure 3) |
|
Patients without esophagitis |
decrease from pretreatment to post-treatment (Figure 3) |
We hypothesis that the improvement in abdominal symptoms in the F scale after eradication might be attributable to improved H. pylori infection related and functional dyspepsia-related symptoms rather than to symptoms by GERD. We explained this in the revised version.
- The statement in line 320 is correct, so why do you argue against your results?
Response
Thank you for your comments.
Reference 12 showed that “heartburn”, not reflex-related symptom combined scores of seven questions by the F-scale in this study, did not decrease from pretreatment to post-treatment in meta-analysis. Because reflex-related symptom has many kinds of symptom including heart burn, we investigated on reflex-related symptom combining scores of several questions in this study. Therefore, we think that we do not argue against our results.